# Combining Four Gaussian Lasers Using Silicon Nitride MMI Slot Waveguide Structure

**DOI:** 10.3390/mi13101680

**Published:** 2022-10-06

**Authors:** Netanel Katash, Salman Khateeb, Dror Malka

**Affiliations:** Faculty of Engineering Holon, Institute of Technology (HIT), Holon 5810201, Israel

**Keywords:** MMI coupler, silicon nitride, slot waveguide, optical combiner

## Abstract

Transceivers that function under a high-speed rate (over 200 Gb/s) need to have more optical power ability to overcome the power losses which is a reason for using a larger RF line connected to a Mach–Zehnder modulator for obtaining high data bitrate communication. One option to solve this problem is to use a complex laser with a power of over 100 milliwatts. However, this option can be complicated for a photonic chip circuit due to the high cost and nonlinear effects, which can increase the system noise. Therefore, we propose a better solution to increase the power level using a 4 × 1 power combiner which is based on multimode interference (MMI) using a silicon nitride (Si_3_N_4_) slot waveguide structure. The combiner was solved using the full-vectorial beam propagation method (FV-BPM), and the key parameters were analyzed using Matlab script codes. Results show that the combiner can function well over the O-band spectrum with high combiner efficiency of at least 98.2% after a short light coupling propagation of 28.78 μm. This new study shows how it is possible to obtain a transverse electric mode solution for four Gaussian coherent sources using Si_3_N_4_ slot waveguide technology. Furthermore, the back reflection (BR) was solved using a finite difference time-domain method, and the result shows a low BR of 40.15 dB. This new technology can be utilized for combining multiple coherent sources that work with a photonic chip at the O-band range.

## 1. Introduction

The rapid expansion of Internet data over the world has increased the need for high-speed communication systems that employ photonic integrated circuit technologies, which enable us to operate with less power loss [1]. Due to the rapidly increasing bandwidth demand from many communication services, such as the Internet, which have developed from web surfing and email to video streaming, social networking, and cloud computing, data centers generate a significant amount of Internet data traffic that uses more transport bandwidth than conventional telecommunication [2]. Thus, scalable optical transceiver technology is needed for the upcoming generations of data centers to handle the predicted exponential increase of data center networks. A major technology to assist with the increase in bandwidth at closer ranges is silicon (Si) photonics, which is based on optical interconnects [3]. Utilizing the current complementary metal–oxide–semiconductor foundries, Si photonics is a low-cost, high-yield manufacturing platform for co-integrating high-performance photonics components [4,5].

Due to the increased demand for data transmission across a variety of applications, high-bitrate modulator devices are most often used. These devices can still function at lower rates, but they lead to higher power losses [6]. To overcome these losses, more laser power is required (over a hundred milliwatts) to achieve minimal power losses. A powerful laser can solve this issue, but it can also degrade the system performance due to its nonlinear effect, which may amplify noise over the channel and is typically more expensive than a simple laser [7]. The usage of a combiner waveguide, which can combine multiple sources into one source, is a superior alternative for a powerful laser. Two high-index material rails (strips) and a low-index region make up a slot waveguide [8,9]. A slot waveguide is a special type of waveguide that can focus and direct light in the slot/slab region under the transverse electric (TE) or transverse magnetic (TM) polarization mode. As a result, the light has strong power confinement and can be guided by the total internal reflection effect. The discontinuity of the electric field at the interfaces of the various materials is another crucial aspect of slot waveguide operation. At the slot boundaries, the electromagnetic wave’s TE mode’s perpendicular electric field component exhibits a discontinuity. Based on Maxwell’s equations, this analysis demonstrates that the electric field is significantly stronger in the low refractive index slot region than in the high refractive index regions. As a result, the power density in the slot is relatively higher than that in the high refractive index regions [9]. Due to these effects, a slot waveguide is an appealing option for waveguide applications, such as modulators [10], polarization beam splitting [11], sensing [12], wavelength conversion [13], demultiplexers [14], and power splitters [15,16] that require low power losses.

The use of MMI devices in integrated optics has received increasing attention in recent years [17,18]. Self-imaging, a feature of multi-mode waveguides, allows an input field profile to be replicated in single or multiple pictures at regular intervals along the propagation path of the guide and serves as the foundation for the MMI primary working theory [8]. Because MMI has outstanding qualities, such as a wide bandwidth and minimal power losses [19,20], photonic circuits heavily rely on MMI that is based on photonic components. Typically, the beam propagation method (BPM) is used to investigate MMI devices [21,22]. Many other applications, including filters [23], temperature sensors [24], splitters [25,26], and couplers [27], make use of MMI devices.

The amount of light reflected from the output waveguide is known as the back reflection (BR). These phenomena can happen because the waveguide materials along the boundaries have different reflecting indices. The BR that returns to the laser source is one of the primary issues that can restrict the performance of the transmitter system [14]. Due to the Si and silica (SiO_2_) MMI coupler’s large index difference, it can acquire reflections from the self-imaging process [14]. However, because of its excellent features [28,29,30], such as a low refractive index, low absorption at the O-band range, and low-temperature sensitivity, a low index material such as silicon nitride (Si_3_N_4_) can be a fantastic answer.

The Si_3_N_4_ waveguide material provides good aspects for designing photonic devices due to its superior properties which are low absorption and low attenuation for a wide range of spectrums including the O-band range. It also has good thermal stability to the operated wavelength drift that can happen due to the laser heat process [31,32]. The index difference between Si and SiO_2_ or Si_3_N_4_ and SiO_2_ significantly influences the waveguide guiding qualities [33]. In SiO_2_ cladded with Si, the index contrast is roughly 145%, and it does not considerably vary with the signal shift. The index contrast in a Si_3_N_4_ platform with a SiO_2_ coating is approximately 35% at the O-band spectrum. The advantages of Si’s better index contrast over the Si_3_N_4_ platform come at the expense of Si devices being more sensitive to geometrical alterations on the order of nanometers. These can result from fabrication faults or differences in the Si layer’s thickness, which affect the waveguide’s width and etch depth. Effective index deviations from the required parameter set a crucial key parameter for the MMI coupler device. Therefore, for an MMI coupler that operates in an O-band window, the Si_3_N_4_ platform is the ideal choice.

A Si slot waveguide power combiner based on MMI technology was presented on [34], reaching a power combining of 97% of the normalized input power with a back reflection (BR) of 33.24 dB. However, this BR value can be problematic for lasers with a limit of 35 dB of BR. This issue can be solved using a complex Si angled MMI design with a significant trade-off of the combiner efficiency of around 10% and more required lithographic process on the chip wafer [34]. To avoid this complex structure, a better way is to use SI_3_N_4_ slot waveguide technology for the realization of the MMI power combiner. Another benefit of using the Si_3_N_4_ slot waveguide structure is the ability to achieve a compact footprint size without obtaining a significant power loss [32].

In this work, we propose a design of a 4 × 1 MMI power combiner based on SI_3_N_4_ slot waveguide technology. Simulation and analysis of key geometrical parameters were implemented using an FV-BPM, aiming for a high-power combining efficiency and strong power confinement inside the SI_3_N_4_ areas. Input/output tapers were optimized and used to reduce losses and BR. The combiner function under a TE fundamental mode has a better performance compared to the TM mode. The suggested combiner can be handy in combining four coherent laser sources operating at a wavelength of 1.31 μm for achieving a higher power level in optical transmitter systems.

## 2. The 4 × 1 Power-Combiner Structure and Theoretical Aspects

The introduced slot waveguide structure as shown in Figure 1a is based on one slot layer and two slab layers. Figure 1a illustrates the slot waveguide structure at the x-y plane, where H_Si3N4_ is the SI_3_N_4_ thickness at the slab area, and H_SiO2_ is the SiO_2_ thickness at the slot area. The green areas are the Si_3_N_4_ slab, and the white areas are the SiO_2_ clad and slot, with a refractive index of 1.994 and 1.447, respectively, at a wavelength of 1310 nm.

Figure 1b illustrates a sketch of the MMI coupler combiner, the input tapers, and output taper structures at the x-z plane. In this sketch, W_MMI_ and L_MMI_ are the width and length of the MMI coupler, respectively, and Gap is the distance between two laser sources.

The basic operation of the MMI coupler is strongly based on the self-imaging effect, and the beat length of the lowest order modes is mentioned in [34]:(1)$$L_{\pi }\approx \frac{4n_{eff}W_{e}{}^{2}}{3\lambda }$$

In our case, the operation wavelength λ is 1310 nm, n_eff_ is the effective refractive index, and W_e_ is the effective width of the MMI coupler which is mentioned in [34]:(2)$$W_{e}=W_{MMI}+\left(\frac{\lambda }{\pi }\right){\left(\frac{n_{clad}}{n_{eff}}\right)}^{2\sigma }\frac{1}{\sqrt{n_{eff}{}^{2}-n_{clad}{}^{2}}}$$
where σ = 0 is suitable for TE mode, and σ = 1 is suitable for TM mode. As shown in Equation (2), it is easy to see that a compact device can be obtained by selecting the TE mode operation. The length of the MMI coupler is mentioned in [34]:(3)$$L_{MMI}=\frac{3pL_{\pi }}{4N}$$

To find the L_MMI_, we need to choose the parameter P to be a positive number, and N is the number of sources at the input of the MMI coupler. So, in our case, N = 4, p = 1.

In slot waveguide structures, the boundary conditions of the electric field between the low refractive index material (SiO_2_), and the high refractive index material (Si_3_N_4_) are the reason for strong light confinement. The amplitude ratio (AR) of the electric field can be calculated in [35]:(4)$$AR=\frac{E_{\left|x\right|=-b}}{E_{\left|x\right|=+b}}=\frac{n_{Si3N4}^{2}}{n_{SiO2}^{2}}$$
where b is the value of the axis at the discontinuity boundary condition, and the different refractive indexes are presented in Figure 1a. Using Equation (4), the AR is found for the operating wavelength of 1310 nm, and its value is 1.898.

To find the insertion loss (IL) of the device, we will need to use the following equation:(5)$$Losses[dB]=-10Log_{10}\left(\frac{P_{out}}{NP_{in}}\right)$$
where P_out_ is the optical power at the output of the component, and P_in_ is the optical power at the input of the component.

## 3. Simulation Results

The simulations of the 4 × 1 MMI power combiner based on Si_3_N_4_ slot waveguide were performed using Rsoft-cad software. The graphs were plotted using Matlab script codes based on the collected data from the FV-BPM results. The thick layer of both the Si_3_N_4_ and SiO_2_ were set to 350 nm and 100 nm, respectively. The thickness of the SiO_2_ layer was optimized according to the slot waveguide physical behavior of the TE fundamental mode field. The normalized power level of the TE fundamental mode was simulated and solved for finding the optimal slot thickness size as shown in Figure 2. From this figure, the optimal value of the SiO_2_ layer can be found and is 100 nm. To satisfy the limit error fabrication that can happen today in a good fab that works with high precision, at least a ±20 nm range was set to be the tolerance range [34]. Therefore, the tolerance range was set between 80–120 nm which is suitable for 98% of the normalized power as shown in Figure 2. The thickness of the Si_3_N_4_ layer was set to be 350 nm to match the standard ability that exists today in a fab facility to fabricate an Si_3_N_4_ layer which is usually between 300–400 nm.

Figure 3a shows the TE fundamental mode field profile inside the Si_3_N_4_ slot waveguide structure of the four coherent sources for the operating wavelength of 1310 nm at the x-y plane. It can be noted that strong power confinement is obtained in the slab areas as shown in Figure 3a in red, as expected from the physical behavior of the slot waveguide structure. In addition, a wide range of 800 nm over the y-axis with a strong power is achieved. This range can be utilized for better obtaining light coupling adjustment between the laser sources and the combiner over the *y*-axis. Figure 3b shows the normalized power for each source over the x-axis at y = 175 nm which is located at the center of the Si_3_N_4_ slot layer.

Figure 4a shows the TM fundamental mode field profile for the four coherent sources for the operation wavelength of 1310 nm at the x-y plane. It can be observed that the light is strongly confined inside the SiO_2_ layer area (red color) as shown in Figure 4a. The normalized light power level for each TM fundamental mode over the *x*-axis at the center of the SiO_2_ layer can be seen in Figure 4b. Considering the results of the simulations shown in Figure 3 and Figure 4, the TM mode has a less normalized power level compared to the TE mode. Furthermore, the TE mode has better flexibility to laser alignment precision over the y-axis with an 800 nm range. Thus, the combiner was chosen to function under the TE mode.

By solving the TE fundamental mode, the n_eff_ value was found which is 1.72. To ensure a compact size device and good stability to fabrication errors or wavelength drift, the MMI coupler width was selected to be 3.5 μm. By using these values in Equations (1) and (2), W_e_ and L_π_ were calculated, resulting in 3.94 μm for We and 28.78 μm for Lπ. Using Equation (3) and FV-BPM simulations, the optimal values of L_MMI_ and W_MMI_ were found, and their values are 9.3 μm and 4.8 μm, respectively.

Figure 5a shows the tolerance range of the width of the MMI coupler, and Figure 5b shows the tolerance range of the length of the MMI coupler. As we can see, the tolerance range is 4.2–5.5 μm (under 97% of the normalized power) and 8.65–9.75 μm (under 98% of the normalized power), respectively. These tolerance values have over 0.45 μm flexibility which is considered good for the fabrication process of the MMI coupler [17].

The input waveguide taper locations were optimized to avoid light coupling between closer laser sources and using FV-BPM simulations. The optimal Gap was found and its value is 0.75 μm. The length of the input taper is 17 μm with a width that varies between 0.5 to 0.9 μm, and the length of the output taper is 2.5 μm with a width that varies between 1 to 0.5 μm. To satisfy the adiabatic condition of the waveguide taper, it is required to conduct optimizations on the input taper length as shown in Figure 6. From this figure, it is clear that the optimal value is 22 μm which fulfills the adiabatic condition. It is also compact.

Figure 7a shows the light propagation of four Gaussian TE mode field laser sources under the operating wavelength of 1.31 μm at the x-z plane. In this simulation, each source was set to a 25% power level. It can be noted that the light combining from four sources to three sources occurs at z = 18 μm, from three sources to two sources occurs at z = 20.5 μm and finally into one source at z = 25 μm. The light combiner efficiency under the TE mode field can reach 98.4% from the input power sources after light propagation of 28.8 μm, as shown in Figure 7b. The total IL is calculated for the operating wavelength using Equation (5), and its value is 0.07 dB.

It is important to emphasize that this solution is suitable for coherent Gaussian sources; in the case of non-coherent sources, an additional component is required, such as a PIN phase-shifter [36] or thermos optic phase-shifter heaters [37] for tuning the phase to resolve the mismatch phase problem between the input sources.

Figure 8 shows the normalized power of the combiner over the O-band spectrum (1.26–1.36 µm), and from this figure, it can be seen that the combiner has a high efficiency of 98.2–98.45% over the O-band spectrum. As can be seen from Figure 8, the combiner was designed to have maximum power at 1310 nm and good stability to the wavelength drift over the O-band range with at least 98.2% of the total power.

The BR that enters the input waveguide taper is an undesirable characteristic due to the potential to cause laser damage. The BR is caused by the self-imaging effect, and due to the high differences between the material’s refractive index on the waveguide boundaries, the level of the power light propagation in the opposite direction can be high.

In this design, we utilize the Si_3_N_4_ slot waveguide and the inputs/output tapers for dramatically reducing the effect of the BR power. Using finite difference time domain (FDTD) simulation as shown in Figure 9a, the BR power value was found which is more than 40 dB. In this simulation, a monitor was placed under the input taper using a segmented waveguide for collecting the reflections coming back from the MMI coupler as shown in Figure 9a. It can be seen that the BR values were more than 39.1 dB over the O-band spectrum. The combiner can function with a very low BR around 40 dB for a wavelength tolerance range of 1300 to 1330nm as shown in Figure 9. In our case, the device was designed and optimized to work at the wavelength of 1310 nm. Thus, as can be observed in Figure 9b, the optimal power BR value is 40.15 dB.

Another important aspect from the particle view is that the laser sources function as a non-coherent Gaussian source. To study the penalty from this aspect, a simulation with non-coherent sources was conducted by inserting a different phase for each Gaussian source. Figure 10 shows the normalized power as a function of the overall phase shift between the four non-coherent sources. As can be seen, our proposed combiner can function well with a high combiner efficiency of 90% and above for phase shifts from 0 to 17.3 degrees. However, the penalty for higher change in the phase can reduce the power combiner level as can be seen in the case of 30 degrees with a combiner efficiency of 70.7%. To resolve this issue, a controller which controls the phase of the four input sources needs to be added to the input waveguide section, and such a device is called a thermo-optic phase-shifter. For our design, the most suitable thermo-optic phase-shifter type is titanium nitride (TiN) metal [38] which can be located above the SI3N4 slot waveguide layers, and the heating of the metal can be controlled by a driver voltage. This solution can be used to match the phase between the input sources. However, it costs more from an energy point of view because of the additional electrical power that is needed for heating the TiN metal.

To understand and study the advantages of this design, a comparison between the proposed combiner design to other combiner devices previously published was completed. As shown in Table 1, the main characteristics of the power combiners were compared including dimension (length × width), number input of sources, operation spectrum band, IL, and BR. It can be seen from Table 1 that the main benefits of the proposed design in this work compared to other combiner devices are the low BR and IL. These advantages can be very useful for integrating this combiner into a transmitter system that operates at the O-band spectrum.

## 4. Conclusions

In this study, we have demonstrated that a Si_3_N_4_ MMI combiner in a slot waveguide structure can be used to achieve a higher power level. The Si_3_N_4_ MMI coupler was demonstrated to combine four coherent Gaussian laser sources with good efficiency at the operating wavelength of 1310 nm under TE mode polarization, across a light propagation of 28.8 μm, using the FV-BPM algorithm. The mode solution reveals that the TE mode has a wide space of 800 nm over the y-axis, which can enhance the light coupling between the input waveguide taper modes and the Gaussian laser source mode. Moreover, this work shows how to use a multi Si_3_N_4_ slot waveguide configuration with a short Gap of 750 nm to obtain a good mode solution with a high intensity level for the four Gaussian laser sources. The device is capable of operating with a high-intensity range of 98.2–98.45% of the total power over the O-band range. As a result, this device has good sensitivity to laser wavelength drift, which can occur in transmitter systems operating in the O-band range due to the heating laser process.

This study demonstrates that employing a laser with a higher power level (more than 100 milliwatts) to compensate for the power losses caused by using a high-speed modulator, such as the Mach–Zehnder Modulator, is one solution. As an alternative, it is preferable to utilize the power combiner that has been suggested because it effectively combines four regular lasers into a single waveguide source that has a greater power level. Another important benefit achieved in this study is that the suggested approach is low-cost compared to the complex laser because this combining technique uses a classical linear laser. Thus, the transmitter system does not have to handle nonlinear effects that may happen from utilizing a laser with a greater level of power and high cost.

The results show that the MMI coupler width and length can adjust for significant changes in the specifications, making it a particularly robust coupler for the fabrication process. Additionally, the utilization of Si_3_N_4_ slot waveguides and taper waveguides to achieve an extremely low BR of 40.15 dB was demonstrated using FDTD simulations. In order to improve channel performances, this study can be used for the combination of four coherent O-band laser sources that are integrated into a high-speed transmitter system. This work can be used to better understand how to combine various coherent or non-coherent laser sources at the O-band window to boost transmitter system power. A phase-shifter or heater can be added to the waveguide input segment to shift the phase and thus can be used to solve the challenging problem with non-coherent sources.

## Figures and Tables

**Figure 1 micromachines-13-01680-f001:**
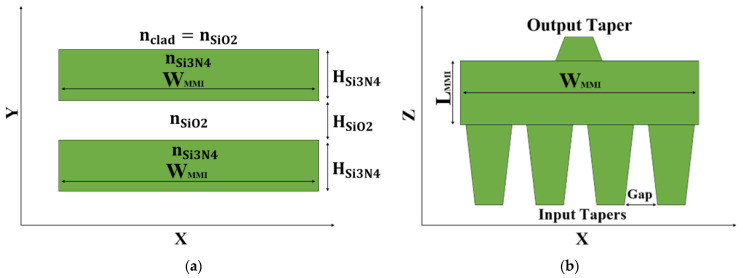
Illustration of the 4 × 1 MMI coupler intensity combiner: (**a**) XY cross-section view. (**b**) XZ cross-section view.

**Figure 2 micromachines-13-01680-f002:**
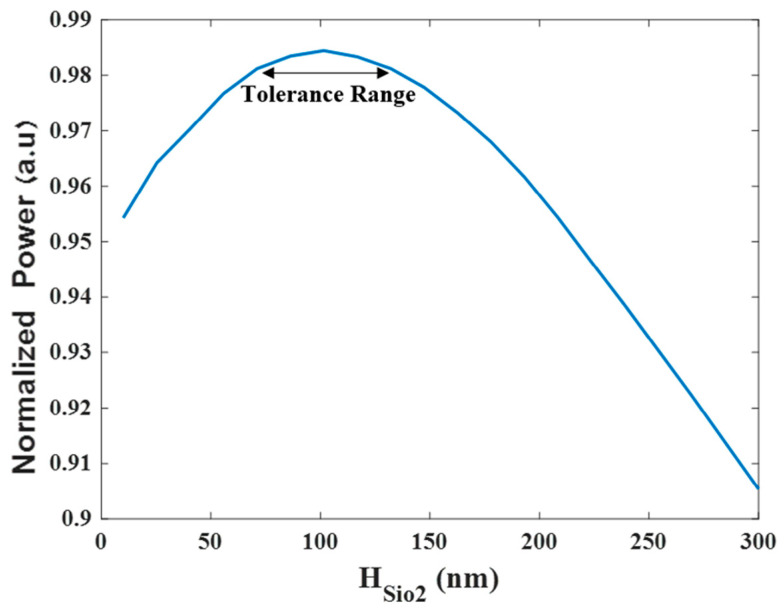
Tolerance analysis and optimizations of H_SiO2_.

**Figure 3 micromachines-13-01680-f003:**
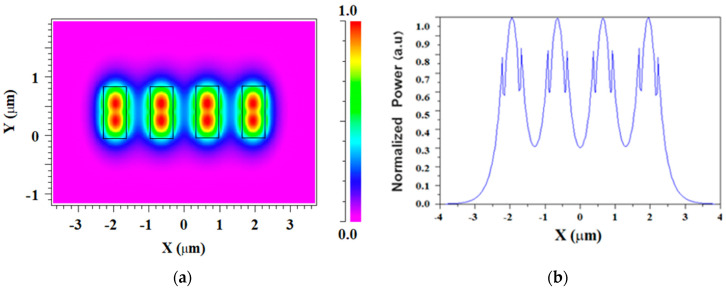
TE fundamental mode solution for four slot waveguide structures: (**a**) E_y_ mode field pattern; (**b**) Horizontal cut at y = 175 nm.

**Figure 4 micromachines-13-01680-f004:**
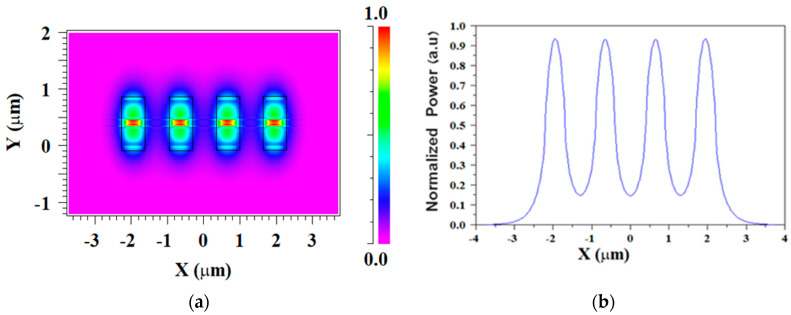
TM fundamental mode solution for four slot waveguide structures: (**a**) E_y_ mode field pattern; (**b**) horizontal cut at y = 400 nm.

**Figure 5 micromachines-13-01680-f005:**
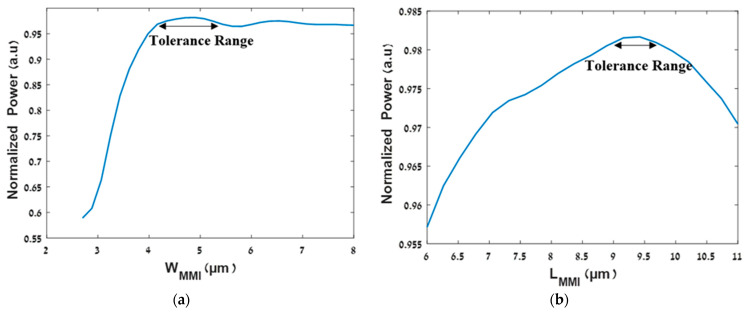
Optimizations of the MMI coupler geometrical parameters: (**a**) normalized power vs. W_MMI_; (**b**) normalized power vs. L_MMI_.

**Figure 6 micromachines-13-01680-f006:**
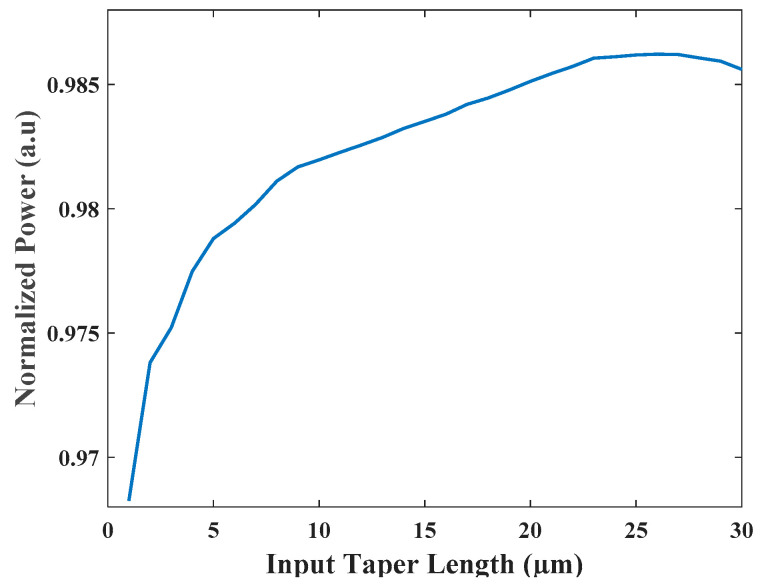
Normalized power as a function of the taper length.

**Figure 7 micromachines-13-01680-f007:**
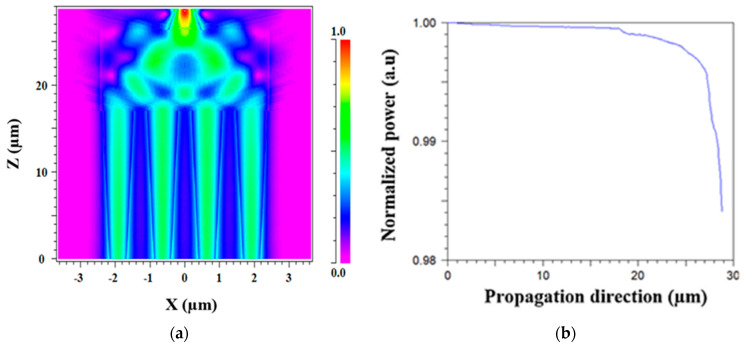
Intensity light propagation profile for the 4 × 1 MMI power combiner: (**a**) plane x-z under TE mode; (**b**) z-axis under TE mode.

**Figure 8 micromachines-13-01680-f008:**
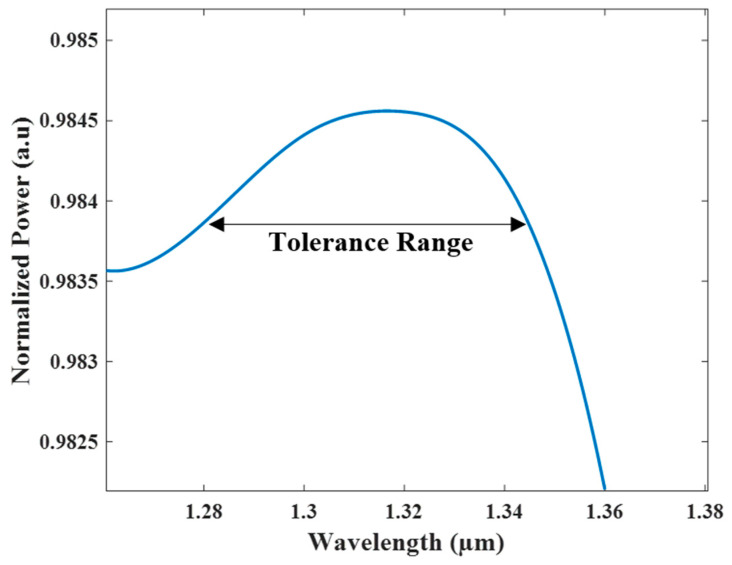
The normalized power of the combiner over the O-band spectrum.

**Figure 9 micromachines-13-01680-f009:**
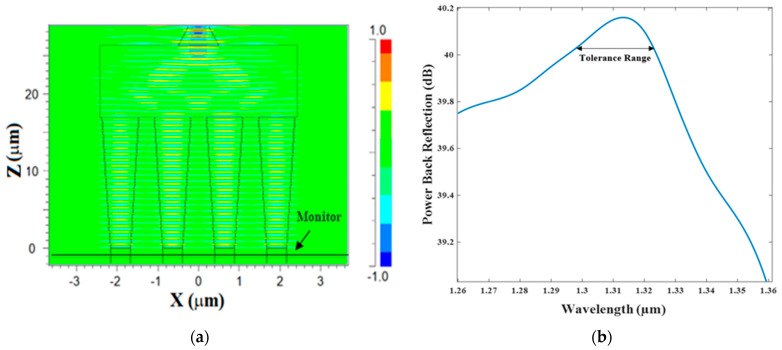
BR calculation using FDTD solver: (**a**) illustration of the simulation setup; (**b**) the power BR over the O-band spectrum.

**Figure 10 micromachines-13-01680-f010:**
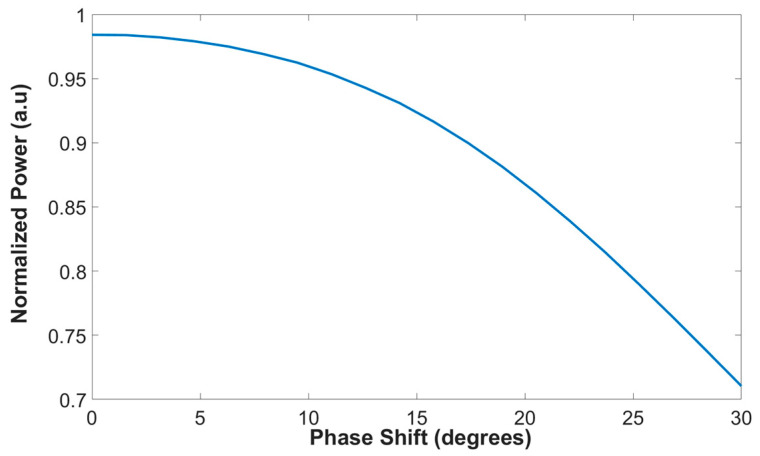
The normalized power as a function of the overall phase shift of the four laser sources.

**Table 1 micromachines-13-01680-t001:** Comparison between the key parameters of different waveguide power combiners.

Waveguide Combiner Type	Dimension (mm) (Length × Width)	Number of Inputs Sources	Operation Spectrum Band	Insertion Loss (dB)	BR (dB)	Year of Publication
MMI using Si slot waveguide [34]	0.00982 × 0.0035	3	C-band	~0.088	33.25	2020
MMI using InP/InGaAsP waveguide [39]	1.72 × 0.006	2	O-band	~0.78	NA	2015
MMI based on Si_3_N_4_ slot waveguide	0.028 × 0.0048	4	O-band	~0.07	40.15	This work

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
