# Peer review of "Combining Four Gaussian Lasers Using Silicon Nitride MMI Slot Waveguide Structure"

_micromachines, 2022, doi:10.3390/mi13101680_

Round 1
Reviewer 1 Report
The manuscript reports an interesting simulation study about slot waveguides MMI combiner. I would say that the main problem in a practical application of this structure wuld be to ensure that the 4 input beams are perfectly in phase. So an additional stage of control should be included in the real life device. Maybe the authors can include a discussion abut that? Eventually also to report some simulation results about what happens if there is some phase shift and up to which level it can be acceptable in terms of power attenuation in the output beam.
A Minor comment: Why in Figure 3 the cut is taken at 175nm instead of the middle value 400nm?
Author Response
Reviewer :
The manuscript reports an interesting simulation study about slot waveguides MMI combiner. I would say that the main problem in a practical application of this structure would be to ensure that the 4 input beams are perfectly in phase. So an additional stage of control should be included in the real life device. Maybe the authors can include a discussion about that? Eventually also to report some simulation results about what happens if there is some phase shift and up to which level it can be acceptable in terms of power attenuation in the output beam.
Answer:
Thank you for your suggestions which are very helpful for improving the quality of the paper. We added new simulation results (figure 10) which show what happens if there is some phase shift and up to study the stability of the proposed device in case of phase changes between the four sources. Also, we include a full discussion about adding a phase-shifter controller as the reviewer suggested.
Reviewer :
- A Minor comment: Why in Figure 3 the cut is taken at 175nm instead of the middle value 400nm?
Answer:
We thank the reviewer for his comment. In figure 4 the cut was made at the middle of the SiO2 layer where the light is strongly confined under the given TM mode, while in figure 3 the cut was made at the middle of the Silicon Nitride layer which is 175nm where the light is strongly confined under TE mode.

Author Response
Reviewer:
In this paper, entitled ‘Combining Four Gaussian Lasers Using Silicon Nitride MMI Slot Waveguide Structure’, the authors numerically design a 4-port beam combiner for the purpose of increasing the light power in Si3N4 integrated platforms. In order to have high transmission and low back reflection, the slot waveguide scheme is applied. In general, the paper is well written, but the simulation and the analysis are not very rigorous in my opinion. Before I could recommend this paper to be accepted, the authors are supposed to address the following concerns:
Si3N4 is an amorphous material and is not stable under high power. In other words, photonic devices made of Si3N4 can be burnt with high input power. Thus, I am just wondering is it necessary or suitable to use the beam combiner in Si3N4. Meanwhile, is the slot waveguide structure compatible with normal SiNOI integrated platform?
Answer:
We thank the reviewer for his comment. Recently researchers stat do design and dealing with silicon nitride waveguides instead of using silicon waveguides because of the low back reflection which can be obtained using low-index material. Thus, in our case we are not using a high-power level, usually, the main challenge in a particle silicon photonics system is to set the laser inside the chip (maybe in a flip way is better) meaning the mode matching will cause at least to 2-3 dB losses and then the high-speed modulator (MZM) can add 10 dB power losses and more. This is why more power is needed (around 100mw), but still is not the level of high-power condition which can cause burnt for the silicon nitride. From theory, each material that has a higher index than air can be suitable for slot-waveguide structure, however, the light confinement can be low in case of a very low index below 1.5 and it is more difficult to use air as the slot material from fabrication view. Thus, SiNOl can be suitable for slot-waveguide structures.
Reviewer:
When high power light is injected, due to the Kerr effect, the refractive index of the material can be changed. How is the device performance affected?
Answer:
We thank the reviewer for his comment. In this case, the Kerr effect cannot happen because we are using 25-30 mW power level for each laser source which is not suitable for high-power levels. Also, usually at the output of the optical transmitter, the power level can reach only a few mW to 10 mW which is a good power lever for data-center applications.
Reviewer:
How did the author determine the tolerance range in Fig2, 5, 8 and 9. It is also suggested to use 1 dB bandwidth.
Answer:
The tolerance range was set in each figure to satisfy the fabrication error parameter which was taken from the fab faculty (Tower company) for each geometrical parameter. In addition, this tolerance range can be very useful for setting the test structures.
Reviewer:
Regarding to Fig3, it seems like a high-order mode, as there are two bright spots. Can the author explain why it is fundamental mode? Is this mode pure TE mode or with some TM polarization? It is also suggested to draw the waveguide profile or boundaries in the mode.
Answer:
This is not a high-order mode because of using slot-waveguide technology and not classical waveguides (rib or strip) which in classical case the mode looks like a high-order mode but for slot-waveguide, this is the field profile in case of fundamental TE mode polarization and in our case, it’s for four laser sources. Thus, this solution shows the TE mode profile for all four sources and not for one source this is why in figure 3 we observed four fundamental solutions for the TE mode. Also, as can be noticed the high-intensity light is in the high-index areas (SiN layers). The mode is only pure TE mode without any TM polarization, as you can see the TM solution is shown in figure 4 for four laser sources. The waveguide boundaries have been added to figures 3(a) and 4(a) as the reviewer suggested.
Reviewer:
The authors claim that considering the compact size, the MMI coupler width is selected to be 3.5 um. Can the author explain more on this point, e.g. how this parameter is calculated, or why it cannot be 3 or 4 um?
Answer:
We thank the reviewer for his comment. To ensure a compact size device and good stability to fabrication errors or wavelength drift. Thus, the MMI coupler width was selected to be 3.5 μm. In the revised manuscript we added this explanation. It cannot be 3um because this size is cannot fulfill the error fabrication of 20nm, thus the MMI coupler will be more sensitive to the variation of the width size and the signal. It can be possible to design the width MMI coupler to 4um, however, selecting this size will increase the footprint size.
Reviewer:
In Fig 7a, there is strong scattering in the combiner part from Z>18 um. How does the scattering come from, and how to eliminate? As far as I am concerned, the TE mode is not stabilized at 29 um, and the power still decreases dramatically. Thus, Z should be extended. And the performance should be analyzed when the mode is stably propagated in the waveguide.
Answer:
Only less than 2% of light is scattering from the MMI coupler as can be seen by the weak blue color which is around 2%. The scattering happens because of the self-imaging effect inside the MMI coupler which can cause the light reflection and scattering. The figure shows the intensity profile and not the TE mode profile. Thus, the light is stable and the power is not decreasing as can be noticed in figure 7(b), only around 2% power losses which is very low.
Reviewer:
At the end of the second paragraph in 1 Introduction, some recent applications of slot waveguides are missing, for example, modulator [1], polarization beam splitting[2], sensing [3], and wavelength conversion [4].
[1] Kovacevic, Goran, et al. "Ultra-high-speed graphene optical modulator design based on tight field confinement in a slot waveguide." Applied Physics Express 11.6 (2018): 065102.
[2] Shi, Xiaodong, et al. "Compact low-birefringence polarization beam splitter using verticaldual-slot waveguides in silicon carbide integrated platforms." Photonics Research 10.1 (2022): A8-A13.
[3] Steglich, Patrick, and K. Y. You. "Silicon-on-insulator slot waveguides: Theory and applications in electro-optics and optical sensing." Emerging Waveguide Technology (2018): 187-210.
[4] Guo, Kai, et al. "Broadband wavelength conversion in a silicon vertical-dual-slot waveguide." Optics Express 25.26 (2017): 32964-32971. Overall, after the above comments are properly addressed, I recommend the publication of this manuscript in Micromachines
Answer:
Thank you for your suggestions which are very helpful for improving the quality of the paper. In the revised manuscript, we added these references [10-13] for showing more applications such as modulator, polarization beam splitting, sensing [3], and wavelength conversion as the reviewer requested.

Round 2
Reviewer 1 Report
Authors have successfuly accomplished the extension required in the first round of review. I suggest to publish the manuscript.
Reviewer 2 Report
Thanks for the response letter. The authors answered satisfactorily to the reviewer's comments.